# Deep Automodulators

**Ari Heljakka**[1,2]    **Yuxin Hou**[1]    **Juho Kannala**[1]    **Arno Solin**[1]
[1]Aalto University    [2]GenMind
{ari.heljakka, yuxin.hou, juho.kannala, arno.solin}@aalto.fi

## Abstract

We introduce a new category of generative autoencoders called automodulators. These networks can faithfully reproduce individual real-world input images like regular autoencoders, but also generate a fused sample from an arbitrary combination of several such images, allowing instantaneous 'style-mixing' and other new applications. An automodulator decouples the data flow of decoder operations from statistical properties thereof and uses the latent vector to modulate the former by the latter, with a principled approach for mutual disentanglement of decoder layers. Prior work has explored similar decoder architecture with GANs, but their focus has been on random sampling. A corresponding autoencoder could operate on real input images. For the first time, we show how to train such a general-purpose model with sharp outputs in high resolution, using novel training techniques, demonstrated on four image data sets. Besides style-mixing, we show state-of-the-art results in autoencoder comparison, and visual image quality nearly indistinguishable from state-of-the-art GANs. We expect the automodulator variants to become a useful building block for image applications and other data domains.

## 1 Introduction

This paper introduces a new category of generative autoencoders for learning representations of image data sets, capable of not only reconstructing real-world input images, but also of arbitrarily combining their latent codes to generate fused images. Fig. 1 illustrates the rationale: The same model can encode input images (far-left), mix their features (middle), generate novel ones (middle), and sample new variants of an image (conditional sampling, far-right). Without discriminator networks, training such an autoencoder for sharp high resolution images is challenging. For the first time, we show a way to achieve this.

Recently, impressive results have been achieved in random image generation (*e.g.*, by GANs [5, 14, 25]). However, in order to manipulate a real input image, an 'encoder' must first infer the correct representation of it. This means simultaneously requiring sufficient output image quality and the ability for reconstruction and feature extraction, which then allow semantic editing. Deep generative autoencoders provide a principled approach for this. Building on the PIONEER autoencoder [18], we proceed to show that modulation of decoder layers by leveraging adaptive instance normalization (AdaIn, [12, 23, 44]) further improves these capabilities. It also yields representations that are less entangled, a property here broadly defined as something that allows for fine and independent control of one semantic (image) sample attribute at a time. Here, the inductive bias is to assume each such attribute to only affect certain scales, allowing disentanglement [33]. Unlike [23], previous GAN-based works on AdaIn [6, 25] have no built-in encoder for new input images.

In a typical autoencoder, input images are encoded into a latent space, and the information of the latent variables is then passed through successive layers of decoding until a reconstructed image has been formed. In our model, the latent vector independently *modulates* the statistics of each layer of the decoder so that the output of layer $n$ is no longer solely determined by the input from layer $n-1$.

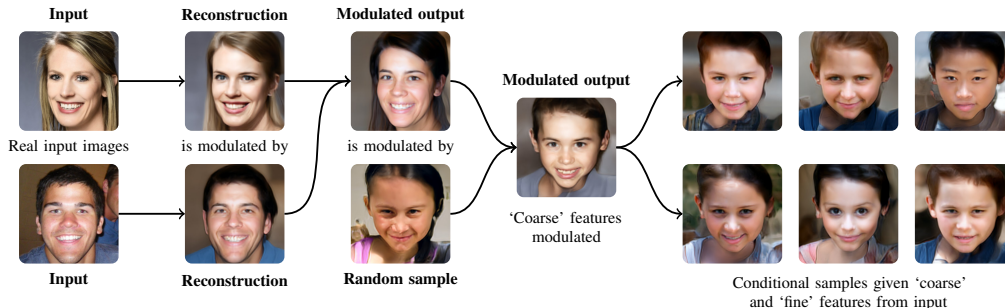

**Input**     **Reconstruction**     **Modulated output**

Real input images     is modulated by     is modulated by

**Modulated output**

'Coarse' features modulated

**Input**     **Reconstruction**     **Random sample**

Conditional samples given 'coarse' and 'fine' features from input

Figure 1: Illustration of some automodulator capabilities. The model can directly encode real (unseen) input images (left). Inputs can be mixed by modulating one with another or with a randomly drawn sample, at desired scales (center); *e.g.*, 'coarse' scales affect pose and gender *etc*. Finally, taking random modulations for certain scales produces novel samples conditioned on the input image (right).

A key idea in our work is to reduce the mutual entanglement of decoder layers. For robustness, the samples once encoded and reconstructed by the autoencoder could be re-introduced to the encoder, repeating the process, and we could require consistency between the passes. In comparison to stochastic models such as VAEs [28, 40], our deterministic model is better suited to take advantage of this. We can take the latent codes of two separate samples, drive certain layers (scales) of the decoder with one and the rest with the other, and then separately measure whether the information contained in each latent is conserved during the full decode–encode cycle. This enforces disentanglement of layer-specific properties, because we can ensure that the latent code introduced to affect only certain scales on the $1^{st}$ pass should not affect the other layers on the $2^{nd}$ pass, either.

In comparison to implicit (GAN) methods, regular image autoencoders such as VAEs tend to have poor output image quality. In contrast, our model *simultaneously* balances sharp image outputs with the capability to encode and arbitrarily mix latent representations of real input images.

The contributions of this paper are as follows. *(i)* We provide techniques for stable *fully unsupervised* training of a high-resolution *automodulator*, a new form of an autoencoder with powerful properties not found in regular autoencoders, including scale-specific style transfer [13]. In contrast to architecturally similar 'style'-based GANs, the automodulator can directly encode and manipulate new inputs. *(ii)* We shift the way of thinking about autoencoders by presenting a novel disentanglement loss that further helps to learn more disentangled representations than regular autoencoders, a principled approach for incorporating scale-specific prior information in training, and a clean scale-specific approach to attribute modification. *(iii)* We demonstrate promising qualitative and quantitative performance and applications on FFHQ, CELEBA-HQ, and LSUN Bedrooms and Cars data sets.

## 2 Related Work

Our work builds upon several lines of previous work in unsupervised representation learning. The most relevant concepts are variational autoencoders (VAEs, [28, 40]) and generative adversarial networks (GANs, [14]). In VAEs, an encoder maps data points to a lower dimensional latent space and a decoder maps the latent representations back to the data space. The model is learnt by minimizing the reconstruction error, under a regularization term that encourages the distribution of latents to match a predefined prior. Latent representations often provide useful features for applications (*e.g.*, image analysis and manipulation), and allow data synthesis by random sampling from the prior. However, with images, the samples are often blurry and not photorealistic, with imperfect reconstructions.

Current state-of-the-art in generative image modeling is represented by GAN models [5, 25, 26] which achieve higher image quality than VAE-based models. Nevertheless, these GANs lack an encoder for obtaining the latent representation for a given image, limiting their usefulness. In some cases, a given image can be semantically mapped to the latent space via generator inversion but this iterative process is prohibitively slow for many applications (see comparison in App. G), and the result may depend on initialization [1, 8].

Bidirectional mapping has been targeted by VAE-GAN hybrids [30, 34–36], and adversarial models [10, 11]. These models learn mappings between the data space and latent space using combinations of encoders, generators, and discriminators. However, even the latest state-of-the-art variant BigBiGAN [9] focuses on random sampling and downstream classification performance, not on faithfulness of reconstructions. InfoGAN [7, 32] uses an encoder to constrain sampling but not for full reconstruction. IntroVAE [22] and Adversarial Generator Encoder (AGE, [45]) *only comprise an encoder and a decoder*, adversarially related. PIONEER scales AGE to high resolutions [17, 18]. VQ-VAE [39, 47] achieves high sample quality with a discrete latent space, but such space cannot, *e.g.*, be interpolated, which hinders semantic image manipulation and prevents direct comparison.

Architecturally, our decoder and use of AdaIn are similar to the recent StyleGAN [25] generator (without the 'mapping network' $f$), but having a built-in encoder instead of the disposable discriminator leads to fundamental differences. AdaIn-based skip connections are different from regular (non-modulating) 1-to-many skip connections from latent space to decoder layers, such as, *e.g.*, in BEGAN [3, 31]. Those skip connections have not been shown to allow 'mixing' multiple latent codes, but merely to map the one and the same code to many layers, for the purpose of improving the reconstruction quality. Besides the AGE-based training [45], we can, *e.g.*, also recirculate style-mixed reconstructions as 'second-pass' inputs to further encourage the independence and disentanglement of emerging styles and conservation of layer-specific information. The biologically motivated recirculation idea is conceptually related to many works, going back to at least 1988 [20]. Utilizing the outputs of the model as inputs for the next iteration has been shown to benefit, *e.g.*, image classification [49], and is used extensively in RNN-based methods [15, 16, 41].

## 3 Methods

We begin with the primary underlying techniques used to construct the automodulator: the progressive growing of the architecture necessary for high-resolution images and the AGE-like adversarial training as combined in the PIONEER [17, 18], but now with an architecturally different decoder to enable 'modulation' by AdaIn [12, 23, 25, 44] (Sec. 3.1). The statistics modulation allows for multiple latent vectors to contribute to the output, which we leverage for an improved unsupervised loss function in Sec. 3.2. We then introduce an optional method for weakly supervised training setup, applicable when there are known scale-specific invariances in the training data itself Sec. 3.3.

### 3.1 Automodulator Components

Our overall scheme starts from unsupervised training of a symmetric convolution–deconvolution autoencoder-like model. Input images $x$ are fed through an encoder $\phi$ to form a low-dimensional latent space representation $z$ (we use $z \in \mathbb{R}^{512}$, normalized to unity). This representation can then be decoded back into an image $\hat{x}$ through a decoder $\theta$.

**Adversarial generator encoder loss**   To utilize adversarial training, the automodulator training builds upon AGE and PIONEER. The encoder $\phi$ and the decoder $\theta$ are trained on separate steps, where $\phi$ attempts to push the latent codes of training images towards a unit Gaussian distribution $N(\mathbf{0}, \mathbf{I})$, and the codes of random generated images away from it. $\theta$ attempts to produce random samples with the opposite goal. In consecutive steps, one optimizes loss $\mathcal{L}_\phi$ and $\mathcal{L}_\theta$ [45], with margin $M_{\text{gap}}$ for $\mathcal{L}_\phi$ [18] (negative KL term of $\mathcal{L}_\theta$ dropped, as customary [17, 46]), defined as

$$\mathcal{L}_\phi = \max(-M_{\text{gap}}, \mathrm{D_{KL}}[q_\phi(z \,|\, x) \,\|\, N(\mathbf{0}, \mathbf{I})] - \mathrm{D_{KL}}[q_\phi(z \,|\, \hat{x}) \,\|\, N(\mathbf{0}, \mathbf{I})]) + \lambda_\mathcal{X} \, d_\mathcal{X}(x, \theta(\phi(x))), \quad (1)$$
$$\mathcal{L}_\theta = \mathrm{D_{KL}}[q_\phi(z \,|\, \hat{x}) \,\|\, N(\mathbf{0}, \mathbf{I})] + \lambda_\mathcal{Z} \, d_{\cos}(z, \phi(\theta(z))), \quad (2)$$

where $x$ is sampled from the training set, $\hat{x} \sim q_\theta(x \,|\, z)$, $z \sim N(\mathbf{0}, \mathbf{I})$, $d_\mathcal{X}$ is L1 or L2 distance, and $d_{\cos}$ is the cosine distance. The KL divergence can be calculated from empirical distributions of $q_\phi(z \,|\, \hat{x})$ and $q_\phi(z \,|\, x)$. Still, the model inference is deterministic, so we could retain, in principle, the full information contained in the image, at every stage of the processing. For any latent vector $z$, decoded back to image space as $\hat{x}$, and re-encoded as a latent $z'$, it is possible and desirable to require that $z$ is as close to $z'$ as possible, yielding the latent reconstruction error $d_{\cos}(z, \phi(\theta(z)))$. We will generalize this term in 3.2.

**Progressively growing autoencoder architecture**   To make the AGE-like training stable in high resolution, we build up the architecture and increase image resolution progressively during training,

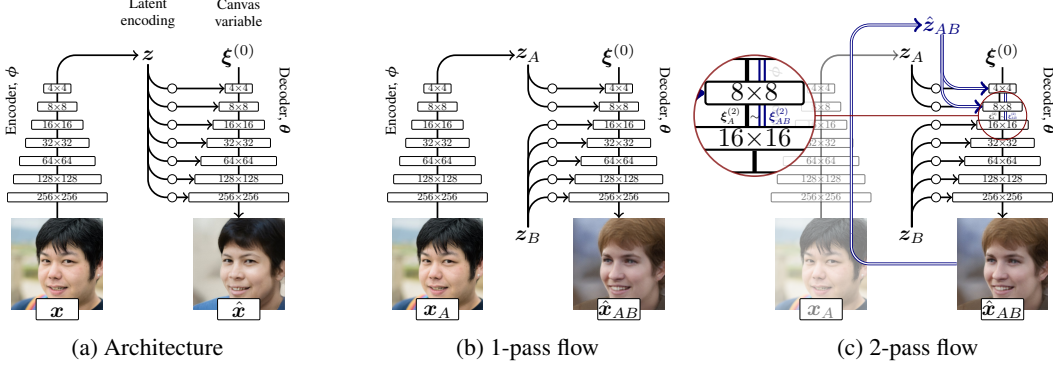

(a) Architecture       (b) 1-pass flow       (c) 2-pass flow

Figure 2: (a) The autoencoder-like usage of the model. (b) Modulations in the decoder can come from different latent vectors. This can be leveraged in feature/style mixing, conditional sampling, and during the model training (first pass). (c) The second pass during training, yielding $\mathcal{L}_j$.

starting from tiny images and gradually growing them, making the learning task harder (see [17, 24] and Supplement Fig. 7). The convolutional layers of the symmetric encoder and decoder are faded in gradually during the training, in tandem with the resolution of training images and generated images (Fig. 7).

**Automodulation** To build a separate pathway for modulation of decoder layer statistics, we need to integrate the AdaIn operation for each layer (following [25]). In order to generate an image, a traditional image decoder would start by mapping the latent code to the first deconvolutional layer to form a small-resolution image ($\boldsymbol{\theta}_0(\boldsymbol{z})$) and expand the image layer by layer ($\boldsymbol{\theta}_1(\boldsymbol{\theta}_0(\boldsymbol{z}))$ *etc.*) until the full image is formed. In contrast, our decoder is composed of layer-wise functions $\boldsymbol{\theta}_i(\boldsymbol{\xi}^{(i-1)}, \boldsymbol{z})$ that separately take a 'canvas' variable $\boldsymbol{\xi}^{(i-1)}$ denoting the output of the preceding layer (see Figs. 2a and 7), and the actual (shared) latent code $\boldsymbol{z}$. First, for each feature map #$j$ of the deconvolutional layer #$i$, we compute the activations $\boldsymbol{\chi}_{ij}$ from $\boldsymbol{\xi}^{(i-1)}$ as in traditional decoders. But now, we modulate (*i.e.*, re-scale) $\boldsymbol{\chi}_{ij}$ into having a new mean $m_{ij}$ and standard deviation $s_{ij}$, based on $\boldsymbol{z}$ (*e.g.*, a block of four layers with 16 channels uses $4 \times 16 \times 2$ scalars). To do this, we need to learn a mapping $\boldsymbol{g}_i : \boldsymbol{z} \mapsto (\boldsymbol{m}_i, \boldsymbol{s}_i)$. We arrive at the AdaIn normalization (also see App. B):

$$\text{AdaIn}(\boldsymbol{\chi}_{ij}, \boldsymbol{g}_i(\boldsymbol{z})) = s_{ij}\left(\frac{\boldsymbol{\chi}_{ij} - \mu(\boldsymbol{\chi}_{ij})}{\sigma(\boldsymbol{\chi}_{ij})}\right) + m_{ij}. \qquad (3)$$

We implement $\boldsymbol{g}_i$ as a fully connected linear layer (in $\boldsymbol{\theta}$), with output size $2\,C_i$ for $C_i$ channels. Layer #1 starts from a constant input $\boldsymbol{\xi}^{(0)} \in \mathbb{R}^{4 \times 4}$. Without loss of generality, here we focus on pyramidal decoders with monotonically increasing resolution and decreasing number of channels.

## 3.2 Conserving Scale-specific Information Over Cycles

We now proceed to generalize the reconstruction losses in a way that specifically benefits from the automodulator architecture. We encourage the latent space to become hierarchically disentangled with respect to the levels of image detail, allowing one to separately retrieve 'coarse' vs. 'fine' aspects of a latent code. This enables, *e.g.*, conditional sampling by fixing the latent code at specific decoder layers, or mixing the scale-specific features of multiple input images—impossible feats for a traditional autoencoder with mutually entangled decoder layers.

First, reinterpret the latent reconstruction error $d_{\cos}(\boldsymbol{z}, \boldsymbol{\phi}(\boldsymbol{\theta}(\boldsymbol{z})))$ in Eq. (2) as 'reconstruction at decoder layer #0'. One can then trivially generalize it to any layer #$i$ of $\boldsymbol{\theta}$ by measuring differences in $\boldsymbol{\xi}^{(i)}$, instead. We simply pick a layer of measurement, record $\boldsymbol{\xi}_1^{(i)}$, pass the sample through a full encoder–decoder cycle, and compare the new $\boldsymbol{\xi}_2^{(i)}$. But now, in the automodulator, different latent codes can be introduced on a per-layer basis, enabling us to measure how much information about a *specific* latent code is conserved at a specific layer, after one more full cycle. Without loss of generality, here we only consider mixtures of two codes. We can present the output of a decoder (Fig. 2b) with $N$ layers, split after the $j^{\text{th}}$ one, as a composition $\hat{\boldsymbol{x}}_{AB} = \boldsymbol{\theta}_{j+1:N}(\boldsymbol{\theta}_{1:j}(\boldsymbol{\xi}^{(0)}, \boldsymbol{z}_A), \boldsymbol{z}_B)$. Crucially, we can choose $\boldsymbol{z}_A \neq \boldsymbol{z}_B$ (extending the method of [25]), such as $\boldsymbol{z}_A = \boldsymbol{\phi}(\boldsymbol{x}_A)$ and

$z_B = \phi(x_B)$ for (image) inputs $x_A \neq x_B$. Because the earlier layers #1:$j$ operate on image content at lower ('coarse') resolutions, the fusion image $\hat{x}_{AB}$ has the 'coarse' features of $z_A$ and the 'fine' features of $z_B$. Now, any $z$ holds feature information at different levels of detail, some empirically known to be mutually independent (*e.g.*, skin color and pose), and we want them separately retrievable, *i.e.*, to keep them 'disentangled' in $z$. Hence, when we *re-encode* $\hat{x}_{AB}$ into $\hat{z}_{AB} = \phi(\hat{x}_{AB})$, then $\theta_{1:j}(\xi^{(0)}, \hat{z}_{AB})$ should extract the same output as $\theta_{1:j}(\xi^{(0)}, z_A)$, unaffected by $z_B$.

This motivates us to minimize the layer disentanglement loss

$$\mathcal{L}_j = d(\theta_{1:j}(\xi^{(0)}, \hat{z}_{AB}), \theta_{1:j}(\xi^{(0)}, z_A)) \qquad (4)$$

for some distance function $d$ (here, L2 norm), with $z_A, z_B \sim \mathrm{N}(\mathbf{0}, \mathbf{I})$, for each $j$. In other words, the fusion image can be encoded into a new latent vector

$$\hat{z}_{AB} \sim q_\phi(z \mid x)\, q_{\theta_{j+1:N}}(x \mid \xi^{(j)}, z_B)\, q_{\theta_{1:j}}(\xi^{(j)} \mid \xi^{(0)}, z_A), \qquad (5)$$

in such a way that, at each layer, the decoder will treat the new code similarly to whichever of the original two separate latent codes was originally used there (see Fig. 2c). For a perfect network, $\mathcal{L}_j$ can be viewed as a 'layer entanglement error'. Randomizing $j$ during the training, we can measure $\mathcal{L}_j$ for any layers of the decoder. A similar loss for the later stage $\theta_{j:N}(\xi^{(j)}, z_B)$ is also possible, but due to more compounded noise and computation cost (longer cycle), was omitted for now.

**Full unsupervised loss** We expect the fusion images to increase the number of outliers during training. To manage this, we replace L1/L2 in Eq. (1) by a robust loss $d_\rho$ [2]. $d_\rho$ generalizes various norms via an explicit parameter vector $\alpha$. Thus, $\mathcal{L}_\phi$ remains as in Eq. (1) but with $d_\mathcal{X} = d_\rho$, and

$$\mathcal{L}_\theta = \mathrm{D_{KL}}[q_\phi(z \mid \hat{x}) \,\|\, \mathrm{N}(\mathbf{0}, \mathbf{I})]$$
$$+ \lambda_\mathcal{Z}\, d_{\cos}(z, \phi(\theta(z))) + \mathcal{L}_j, \quad (6)$$

where $\hat{x}_{1:\frac{3}{4}M} \sim q_\theta(x \mid z)$ with $z \sim \mathrm{N}(\mathbf{0}, \mathbf{I})$, and $\hat{x}_{\frac{3}{4}M:M} \sim q_\theta(x \mid \hat{z}_{AB})$, with a set 3:4 ratio of regular and mixed samples for batch size $M$, $j \sim \mathrm{U}\{1, N\}$, and $\hat{z}_{AB}$ from Eq. (5). Margin $M_{\mathrm{gap}} = 0.5$, except for CELEBA-HQ and Bedrooms 128×128 ($M_{\mathrm{gap}} = 0.2$) and CELEBA-HQ 256×256 ($M_{\mathrm{gap}} = 0.4$). To avoid discontinuities in $\alpha$, we introduce a progressively-growing variation of $d_\rho$, where we first learn the $\alpha$ in the lowest resolution (*e.g.*, 4×4). There, each $\alpha_i$ corresponds to one pixel $p_{x,y}$. Then, whenever doubling the resolution, we initialize the new—now four times as large—$\alpha$ in the higher resolution by replicating each $\alpha_i$ to cover the new $\alpha_j^{1\times 4}$ that now corresponds to $p_{x,y}$, $p_{x+1,y}$, $p_{x,y+1}$ and $p_{x+1,y+1}$, in the higher resolution.

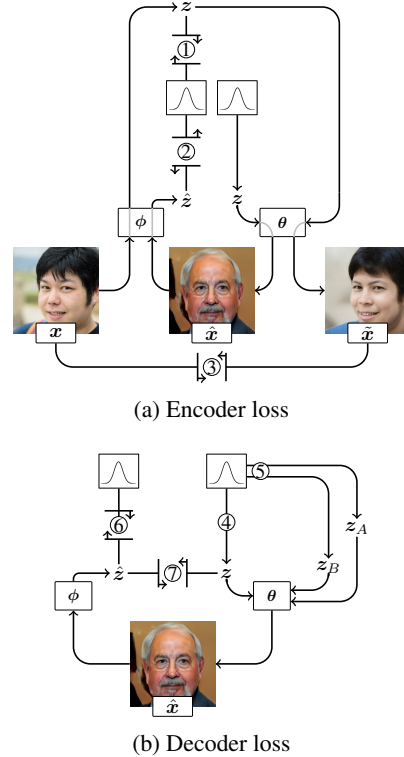

(a) Encoder loss

(b) Decoder loss

Figure 3: Breakdown of the 1-pass flow loss terms.

We summarize the final loss computation as follows. At the encoder training step (Fig. 3a)[45], we compute $\mathcal{L}_\phi$ by first encoding training samples $x$ into latents $z$, minimizing the KL divergence between the distribution of $z$ and $\mathrm{N}(\mathbf{0}, \mathbf{I})$ ①. Simultaneously, we encode randomly generated samples $\hat{x}$ into $\hat{z}$, maximizing their corresponding divergence from $\mathrm{N}(\mathbf{0}, \mathbf{I})$ ②. We also decode each $z$ into $\tilde{x}$, with the reconstruction error $d_\mathcal{X}(x, \tilde{x})$ ③. At the decoder training step, we first compute the 1-pass terms of $\mathcal{L}_\theta$ (Fig. 3b) by generating random samples $\hat{x}$, each decoded from either a single $z$ ④ or a mixture pair $(z_A, z_B)$ ⑤ drawn from $\mathrm{N}(\mathbf{0}, \mathbf{I})$. We encode each $\hat{x}$ into $\hat{z}$ and minimize the KL divergence between their distribution and $\mathrm{N}(\mathbf{0}, \mathbf{I})$ ⑥. We compute the latent reconstruction error $d_{\cos}$ between each $z$ and its re-encoded counterpart $\hat{z}$ ⑦. Finally, for $(z_A, z_B)$, we do the second pass, adding the term $\mathcal{L}_j$ (see Fig. 2c).

## 3.3 Enforcing Known Invariances at Specific Layers

As an extension to the main approach described so far, one can independently consider the following. The architecture and the cyclic training method also allow for a novel principled approach to leverage

known scale-specific invariances in training data. Assume that images $x_1$ and $x_2$ have identical characteristics at some scales, but differ on others, with this information further encoded into $z_1$ and $z_2$, correspondingly. In the automodulator, we could try to have the shared information affect only the decoder layers #$j$:$k$. For any $\boldsymbol{\xi}^{(j-1)}$, we then must have $\theta_{j:k}(\boldsymbol{\xi}^{(j-1)}, z_1) = \theta_{j:k}(\boldsymbol{\xi}^{(j-1)}, z_2)$. Assume that it is possible to represent the rest of the information in the images of that data set in layers #$1$:$(j-1)$ and #$(k+1)$:$N$. This situation occurs, *e.g.*, when two images are known to differ only in high-frequency properties, representable in the 'fine' layers. By mutual independence of layers, our goal is to have $z_1$ and $z_2$ interchangeable at the middle:

$$\boldsymbol{\theta}_{k+1:N}(\boldsymbol{\theta}_{j:k}(\boldsymbol{\theta}_{1:j-1}(\boldsymbol{\xi}^{(0)}, z_2), z_1), z_2) = \boldsymbol{\theta}_{k+1:N}(\boldsymbol{\theta}_{j:k}(\boldsymbol{\theta}_{1:j-1}(\boldsymbol{\xi}^{(0)}, z_2), z_2), z_2)$$
$$= \boldsymbol{\theta}_{1:N}(\boldsymbol{\xi}^{(0)}, z_2) = \boldsymbol{\theta}(\boldsymbol{\phi}(\boldsymbol{x}_2)), \tag{7}$$

which turns into the optimization target (for some distance function $d$)

$$d(\boldsymbol{\theta}(\boldsymbol{\phi}(\boldsymbol{x}_2)), \boldsymbol{\theta}_{k+1:N}(\boldsymbol{\theta}_{j:k}(\boldsymbol{\theta}_{1:j-1}(\boldsymbol{\xi}^{(0)}, z_2), z_1), z_2)). \tag{8}$$

By construction of $\boldsymbol{\phi}$ and $\boldsymbol{\theta}$, this is equivalent to directly minimizing

$$\mathcal{L}_{\mathrm{inv}} = d(\boldsymbol{x}_2, \boldsymbol{\theta}_{k+1:N}(\boldsymbol{\theta}_{j:k}(\boldsymbol{\theta}_{1:j-1}(\boldsymbol{\xi}^{(0)}, z_2), z_1), z_2)), \tag{9}$$

where $z_1 = \boldsymbol{\phi}(\boldsymbol{x}_1)$ and $z_2 = \boldsymbol{\phi}(\boldsymbol{x}_2)$. By symmetry, the complement term $\mathcal{L}'_{\mathrm{inv}}$ can be constructed by swapping $z_1$ with $z_2$ and $x_1$ with $x_2$. For each known invariant pair $\boldsymbol{x}_1$ and $\boldsymbol{x}_2$ of the minibatch, you can now add the terms $\mathcal{L}_{\mathrm{inv}} + \mathcal{L}'_{\mathrm{inv}}$ to $\mathcal{L}_{\phi}$ of Eq. (6). Note that in the case of $z_1 = z_2$, $\mathcal{L}_{\mathrm{inv}}$ reduces to the regular sample reconstruction loss, revealing our formulation as a generalization thereof.

As we push the invariant information to layers #$j$:$k$, and the other information *away* from them, there are less layers available for the rest of the image information. Thus, we may need to add extra layers to retain the overall decoder capacity. Note that in a pyramidal deconvolutional stack where the resolution increases monotonically, if the layers #$j$:$k$ span more than two consecutive levels of detail, the scales in-between cannot be extended in that manner.

# 4   Experiments

Since automodulators offer more applications than either typical autoencoders or GANs without an encoder, we strive for reasonable performance across experiments, rather than beating any specific metric. (Experiment details in App. A.) Any generative model can be evaluated in terms of sample quality and diversity. To measure them, we use Fréchet inception distance (FID) [19], which is comparable across models when sample size is fixed [4], though notably uninformative about the ratio of precision and recall [29]. Encoder–decoder models can further be evaluated in terms of their ability to reconstruct new test inputs, which underlies their ability to perform more interesting applications such as latent space interpolation and, in our case, mixing of latent codes. For a similarity metric between original and reconstructed face images (center-cropped), we use LPIPS [50], a metric with better correspondence to human evaluation than, *e.g.*, traditional L2 norm.

The degree of latent space disentanglement is often considered the key property of a latent variable model. Qualitatively, it is the necessary condition for, *e.g.*, style mixing capabilities. Quantitatively, one could expect that, for a constant-length step in the latent space, the less entangled the model, the smaller is the overall perceptual change. The extent of this change, measured by LPIPS, is the basis of measuring disentanglement as Perceptual Path Length (PPL) [25].

We justify our choice of a loss function in Eq. (6), compare to baselines on relevant measures, demonstrate the style-mixing capabilities specific to automodulators, and show a proof-of-concept for leveraging scale-specific invariances (see Sec. 3.3). In the following, we use Eqs. (1) and (6), and polish the output by adding a source of unit Gaussian noise with a learnable scaling factor before the activation in each decoder layer, as in StyleGAN [25], also improving FID.

Table 1: Effect of loss terms on CELEBA-HQ at $256\times256$ with 40M seen samples (50k FID batch) before applying layer noise.

|  | FID | FID (mix) | PPL |
| --- | --- | --- | --- |
| Automodulator architecture | 45.25 | 52.83 | 206.3 |
| + Loss $\mathcal{L}_j$ | 44.06 | 47.74 | 210.0 |
| + Loss $d_\rho$ replacing L1 | **36.20** | 43.53 | 217.3 |
| + Loss $\mathcal{L}_j + d_\rho$ replacing L1 | 37.95 | **40.90** | **201.8** |

**Ablation study for the loss metric**   In Table 1, we illustrate the contribution of the layer disentanglement loss $\mathcal{L}_j$ and the robust loss $d_\rho$ on the FID for regular and mixed samples from the model

at $256\times256$ resolution, as well as PPL. We train the model variants on CELEBA-HQ [24] data set to convergence (40M seen samples) and choose the best of three restarts with different random seeds. Our hypothesis was that $\mathcal{L}_j$ improves the FID of mixed samples and that replacing L1 sample reconstruction loss with $d_\rho$ improves FID further and makes training more stable. The results confirm this. Given the improvement from $d_\rho$ also for the mixed samples, we separately tested the effect of $d_\rho$ without $\mathcal{L}_j$ and find that it produces even slightly better FID for regular samples but then considerably worse FID for the mixed ones, due to, presumably, more mutually entangled layers. For ablation of the $M_{\text{gap}}$ term, see [18]. The effect of the term $d_{\text{cos}}$ was studied in [45] (for *cos* instead of L2, see [46]).

**Encoding, decoding, and random sampling**     To compare encoding, decoding, and random sampling performance, autoencoders are more appropriate baselines than GANs without an encoder, since the latter tend to have higher quality samples, but are more limited since they cannot manipulate real input samples. However, we do also require reasonable sampling performance from our model, and hence separately compare to non-autoencoders. In Table 2a, we compare to autoencoders: Balanced PIONEER [18], a vanilla VAE, and a more recent Wasserstein Autoencoder (WAE) [43]. We train on $128\times128$ CELEBA-HQ, with our proposed architecture ('AdaIn') and the regular one ('classic'). We measure LPIPS, FID (50k batch of generated samples compared to training samples, STD over 3 runs $< 1$ for all models) and PPL. Our method has the best LPIPS and PPL.

In Table 2b, we compare to non-autoencoders: StyleGAN, Progressively Growing GAN (PGGAN) [24], and GLOW [27]. To show that our model can reasonably perform for many data sets, we train at $256\times256$ on CELEBA-HQ, FFHQ [25], LSUN Bedrooms and LSUN Cars [48]. We measure PPL and FID (uncurated samples in Fig. 4 (right), STD of FID over 3 runs $< .1$). The performance of the automodulator is comparable to the Balanced PIONEER on most data sets. GANs have clearly best FID results on all data sets (NB: a hyper-parameter search with various schemes was used in [25] to achieve their high PPL values). We train on the actual 60k training set of FFHQ only (StyleGAN trained on all 70k images). We also tested what will happen if we try invert the StyleGAN by finding a latent code for an image by an optimization process. Though this can be done, the inference is over 1000 times slower to meet and exceed the automodulator LPIPS score (see App. G and Fig. 16). We also evaluate the 4-way image interpolation capabilities in unseen FFHQ test images (Fig. 13 in the supplement) and observe smooth transitions. Note that in GANs without an encoder, one can only interpolate between the codes of *random* samples, revealing little about the recall ability of the model.

**Style mixing**     The key benefit of the automodulators over regular autoencoders is the style-mixing capability (Fig. 2b), and the key benefit over style-based GANs is that 'real' unseen test images can be instantly style-mixed. We demonstrate both in Fig. 4. For comparison with prior work, we use the randomly generated source images from the StyleGAN paper [25]. Importantly, for our model, they appear as unseen 'real' test images. Performance in mixing real-world images is similar (Supplementary Figs. 14 and 15). In Fig. 4, we mix specific input faces (from source A and B) so that the 'coarse' (latent resolutions $4\times4 - 8\times8$), 'intermediate' ($16\times16 - 32\times32$) or 'fine' ($64\times64 - 512\times512$) layers of the decoder use one input, and the rest of the layers use the other.

**Invariances in a weakly supervised setup**     In order to leverage the method of Sec. 3.2, one needs image data that contains pairs or sets that share a scale-specific prominent invariant feature (or, conversely, are identical in every other respect except that feature). To this end, we demonstrate a

Table 2: Performance in CELEBA-HQ (CAHQ), FFHQ, and LSUN Bedrooms and Cars. We measure LPIPS, Fréchet Inception Distance (FID), and perceptual path length (PPL). Resolution is $256\times256$, except *$128\times128$. For all numbers, **smaller is better**. Only the '-AdaIn' architectures are functionally equivalent to the automodulator (encoding and latent mixing). GANs in gray.

(a) Encoder–decoder comparison

| | LPIPS (CAHQ*) | FID (CAHQ*) | PPL (CAHQ*) |
|---|---|---|---|
| B-PIONEER | 0.092 | **19.61** | 92.8 |
| WAE-AdaIn | 0.165 | 99.81 | **62.2** |
| WAE-classic | 0.162 | 112.06 | 236.8 |
| VAE-AdaIn | 0.267 | 114.05 | 83.5 |
| VAE-classic | 0.291 | 173.81 | 71.7 |
| Automodulator | **0.083** | 27.00 | **62.3** |

(b) Generative models comparison

| | FID (CAHQ) | FID (FFHQ) | FID (Bedrooms) | FID (Cars) | PPL (CAHQ) | PPL (FFHQ) |
|---|---|---|---|---|---|---|
| StyleGAN | **5.17** | 4.68 | **2.65** | 3.23 | 179.8 | 234.0 |
| StyleGAN2 | — | **3.11** | — | **5.64** | — | **129.4** |
| PGGAN | 7.79 | 8.04 | 8.34 | 8.36 | 229.2 | 412.0 |
| GLOW | 68.93 | — | — | — | 219.6 | — |
| B-PIONEER | 25.25 | 61.35 | 21.52 | 42.81 | **146.2** | 160.0 |
| Automodulator | 29.13 | 31.64 | 25.53 | 19.82 | 203.8 | 250.2 |

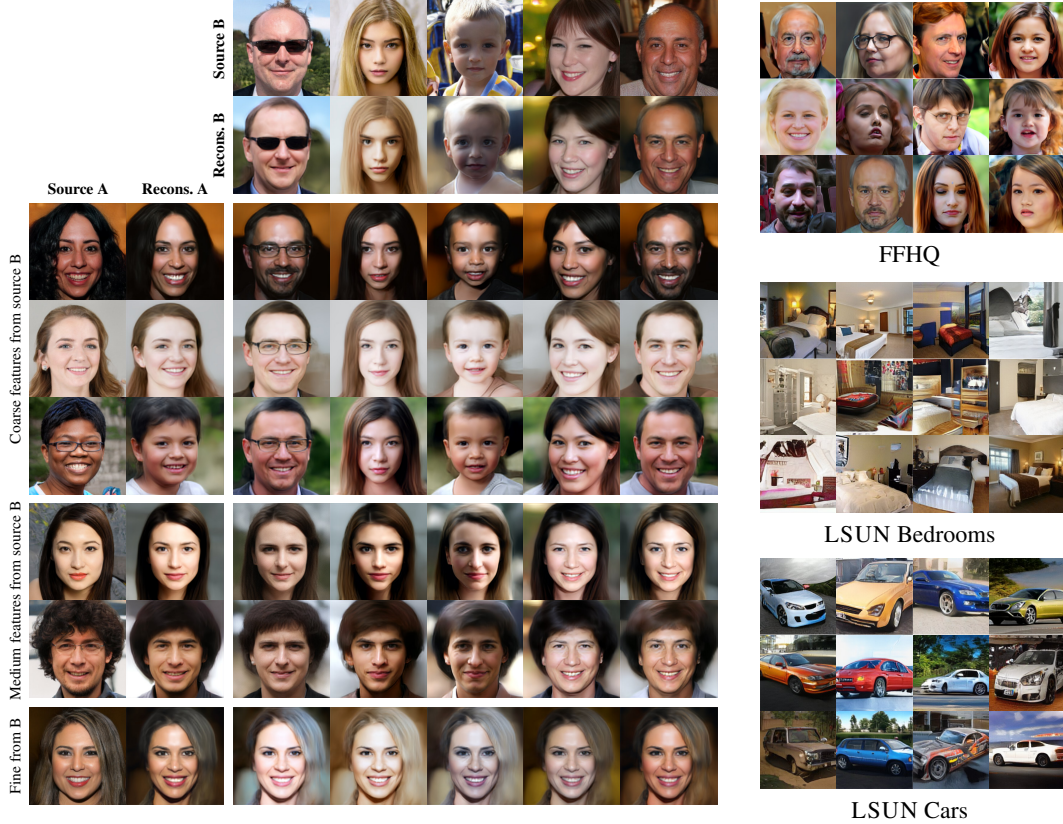

Figure 4: (Left): Feeding the random fake source images in Karras et al. [25] into our model as 'real' inputs, reconstructing at $512{\times}512$ and mixing at three scales. (The same for real faces, see Supplement.) (Right): Uncurated random samples of $512{\times}512$ FFHQ and $256{\times}256$ LSUN.

proof-of-concept experiment that uses the simplest image transformation possible: horizontal flipping. For CELEBA-HQ, this yields pairs of images that share every other property except the azimuth rotation angle of the face, making the face identity invariant amongst each pair. Since the original rotation of faces in the set varies, the flip-augmented data set contains faces rotated across a wide continuum of angles. For further simplicity, we make an artificially strong hypothesis that the 2D projected face shape is the only relevant feature at $4{\times}4$ scale and does not need to affect scales finer than $8{\times}8$. This lets us enforce the $\mathcal{L}_{\text{inv}}$ loss for layers #1–2. Since we do not want to restrict the scale $8{\times}8$ for the shape features alone, we add an extra $8{\times}8$ layer after layer #2 of the regular stack, so that layers #2–3 both operate at $8{\times}8$, layer #4 only at $16{\times}16$, *etc.* Now, with $z_2$ that corresponds to the horizontally flipped counterpart of $z_1$, we have $\boldsymbol{\theta}_{3:N}(\boldsymbol{\xi}^{(2)}, z_1) = \boldsymbol{\theta}_{3:N}(\boldsymbol{\xi}^{(2)}, z_2)$. Our choices amount to $j = 3, k = N$, allowing us to drop the outermost part of Eq. (9). Hence, our additional encoder loss terms are

$$\mathcal{L}_{\text{inv}} = d(\boldsymbol{x}_2, \boldsymbol{\theta}_{3:N}(\boldsymbol{\theta}_{1:2}(\boldsymbol{\xi}^{(0)}, z_2), z_1)) \quad \text{and} \tag{10}$$

$$\mathcal{L}'_{\text{inv}} = d(\boldsymbol{x}_1, \boldsymbol{\theta}_{3:N}(\boldsymbol{\theta}_{1:2}(\boldsymbol{\xi}^{(0)}, z_1), z_2)). \tag{11}$$

Fig. 5a shows the results after training with the new loss (50% of the training samples flipped in each minibatch). With the invariance enforcement, the model forces decoder layers #1–2 to only affect the pose. We generate images by driving those layers with faces at different poses, while modulating the rest of the layers with the face whose identity we seek to conserve. The resulting face variations now only differ in terms of pose, unlike in regular automodulator training.

**Scale-specific attribute editing** Consider the mean difference in latent codes of images that display or do not display an attribute of interest (*e.g.*, smile). Appropriately scaled, such codes can added to any latent code to modify that attribute. Here, one can restrict the effect of the latent code only to the layers driving the expected scale of the attribute (*e.g.*, $16{\times}16 - 32{\times}32$), yielding precise manipulation (App. B, comparisons in Supplement) with only a few exemplars (*e.g.*, [18] used 32).

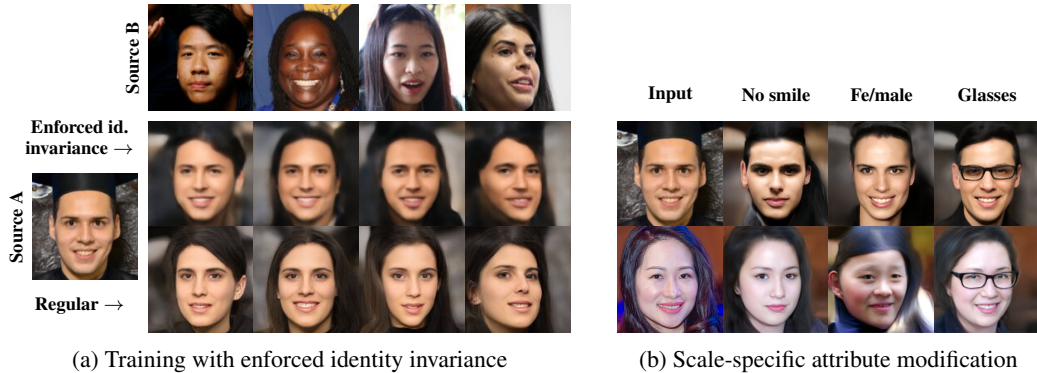

(a) Training with enforced identity invariance     (b) Scale-specific attribute modification

Figure 5: Examples of controlling individual decoder layer ranges at training time and at evaluation time. (a) Training with face identity invariance enforcement under azimuth rotation. We generate images with the 'non-coarse' styles of source A and the 'coarse' ones from each top row image. With 'Enforced identity invariance', the top row only drives the face pose while conserving identity. In comparison, the 'Regular' training lets the top row also affect other characteristics, including identity. (b) Modifying an attribute in latent space by using only 4 exemplar images of it. In 'regular' all-scales manipulation, the variance in the exemplars causes unwanted changes in, *e.g.*, texture and pose. When the latent vector only drives the relevant scales, the variance in other scales is inconsequential.

## 5    Discussion and Conclusion

In this paper, we proposed a new generative autoencoder model with a latent representation that independently modulates each decoder layer. The model supports reconstruction and style-mixing of real images, scale-specific editing and sampling. Despite the extra skill, the model still largely outperforms or matches other generative autoencoders in terms of latent space disentanglement, faithfulness of reconstructions, and sample quality. We use the term *automodulator* to denote any autoencoder that uses the latent code only to modulate the statistics of the information flow through the layers of the decoder. This could also include, *e.g.*, 3D or graph convolutions.

Various improvements to the model are possible. The mixture outputs still show occasional artifacts, indicating that the factors of variation have not been perfectly disentangled. Also, while the layer-induced noise helps training, using it in evaluation to add texture details would often reduce output quality. Also, to enable even more general utility of the model, the performance could be measured on auxiliary downstream tasks such as classification.

Potential future applications include introducing completely interchangeable 'plugin' layers or modules in the decoder, trained afterwards on top of the pretrained base automodulator, leveraging the mutual independence of the layers. The affine maps themselves could also be re-used across domains, potentially offering mixing of different domains. Such examples highlight that the range of applications of our model is far wider than the initial ones shown here, making the automodulators a viable alternative to state-of-the-art autoencoders and GANs.

Our source code is available at https://github.com/AaltoVision/automodulator.

## Broader Impact

The presented line of work intends to shift the focus of generative models from random sample generation towards controlled semantic editing of existing inputs. In essence, the ultimate goal is to offer 'knobs' that allow content editing based on high-level features, and retrieving and combining desired characteristics based on examples. While we only consider images, the techniques can be extended to other data domains such as graphs and 3D structures.

Ultimately, such research could reduce very complex design tasks into approachable ones and thus reduce dependency on experts. For instance, contrast an expert user of a photo editor or design software, carefully tuning details, with a layperson who simply finds images or designs with the desired characteristics and guiding the smart editor to selectively combine them.

Leveling the playing field in such tasks will empower larger numbers of people to contribute to design, engineering and science, while also multiplying the effectiveness of the experts. The downside of such empowerment will, of course, include the threats of deepfakes and spread of misinformation. Fortunately, public awareness of these abuses has been increasing rapidly. We attempt to convey the productive prospects of these technologies by also including image data sets with cars and bedrooms, while comparison with prior work motivates the focus on face images.

## Acknowledgments and Disclosure of Funding

The authors wish to acknowledge the Aalto Science-IT project and CSC – IT Center for Science, Finland, for computational resources. Authors acknowledge funding from GenMind Ltd. This research was supported by the Academy of Finland grants 308640, 324345, 277685, 295081, and 309902. We thank Jaakko Lehtinen and Janne Hellsten (NVIDIA) for the StyleGAN latent space projection script (for the baseline, only) and advice on its usage. We also thank Christabella Irwanto, Tuomas Kynkäänniemi, and Paul Chang for comments on the manuscript.

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
