[Supplementary Material]

# Supplementary Material for
# Deep Automodulators

In the appendix, we include further details underlying the model and the experiments and complement the results in the main paper with examples and more comprehensive results. We start with the details of training and evaluation (App. A), complemented by detailed description of architecture and data flow in the network (App. B). We then show a comparison of scale-specific attribute modification with the regular one, providing more context to the quick qualitative experiment in Sec. 4. We proceed with showing more random samples (App. D) and reconstructions (App. E). Note that there are reconstructions in the diagonals of all style-mixture images, too. Importantly, we show systematic style-mixture examples in App. F, corresponding to Fig. 4 but with *real* (unseen) input images from the FFHQ test set. We follow with showing latent space interpolations at all scales between real input images (which also could be done on a scale-specific basis). We then continue with an experiment regarding the inversion of StyleGAN Karras et al. [25] with an optimization process, and finish with an experiment focused on conditional sampling, in which certain scales of an input image are fixed in the reconstruction images but other scales are randomly sampled over, creating variations of the same input face.

## A    Training Details

The training method largely follows the Balanced PIONEER [18], with progressively growing encoder and decoder with symmetric high-level structure (Fig. 6a), and decreasing the batch size when moving to higher resolutions (Fig. 7). The encoder and decoder consist of 7 blocks each containing two residual blocks with a $3\times3$ filter. In both blocks of the encoder, the convolutions are followed by a spectral normalization operation [37] and activation (conv - spectral norm - act). In the first block of the decoder, they are followed by binomial filtering, layer noise, activation and AdaIn normalization

(a) Top-level          (b) Decoder block structure          (c) Latent vector to modulation

Figure 6: (a) Top-level view, where a single decoder block corresponds to a specific resolution. (b) A single decoder block contains two convolutional layers and other repeating components. The noise is added to each channel of the layer using a single scale parameter per layer, *i.e.*, a different random value is added across each activation map, but the scale is the same for all maps in the same layer. (c) The latent codes are connected to the modulation scalar $(m,s)$ pair of every activation map of each of the $7\times2$ convolutional layers. In the encoder, the number of channels in the convolutional blocks follows [18] as 64,128,256,512,512,512,512. In $256\times256$ CELEBA-HQ, the decoder channels are the symmetric inverse: 512, 512, 512, 512, 256, 128, 64. In other datasets, it was beneficial to double the number of feature maps in high resolutions of the decoder, with the number of channels as: 512, 512, 512, 512, 512, 256, 128 for $256\times256$ datasets. For $512\times512$ FFHQ a final 64-channel block was added to this, resulting in 512, 512, 512, 512, 256, 128, 64.

Figure 7: The model grows step-wise during training; the resolution doubles on every step. Input $x$ is encoded into a latent encoding $z$ (a dimensionality of 512 used throughout this paper). The decoder acts by modulating an empty canvas $\xi^{(0)}$ by the latent encoding and produces the output $\hat{x}$. Further explanation of the model architecture is provided in Fig. 2a.

(conv - binomial - noise - act - AdaIn); in the second block of decoder, by layer noise, activation and AdaIn normalization (conv - noise - act - AdaIn). A leaky ReLU ($p = 0.2$) is used as the activation. Equalized learning rate [24] is used for decoder convolutions. In the encoder, each block halves the resolution of the convolution map, while in the decoder, each block doubles it. The output of the final encoder layers is flattened into a 512-dimensional latent block. As in StyleGAN [25], the block is mapped by affine mapping layers so that each convolutional layer $C$ in the decoder block is preceded by its own fully connected layer that maps the latent to two vectors each of length $N$, when $N$ equals the number of channels in $C$.

Each resolution phase until $32{\times}32$, for all data sets, uses a learning rate $\alpha = 0.0005$ and thereafter 0.001. Optimization is done with ADAM ($\beta_1 = 0, \beta_2 = 0.99, \epsilon = 10^{-8}$). KL margin is 2.0 for the first two resolution steps, and therafter fixed to 0.5, except for CELEBA-HQ, for which it is switched to 0.2 at $128{\times}128$ and 0.4 at $256{\times}256$, and for LSUN Bedrooms, for which the margin was 0.2 from $128{\times}128$ upwards. We believe that 0.5 for low resolutions and 0.2 thereafter would work sufficiently across all these data sets. Note that unlike in [18], we use only one generator training step for each individual encoder training step. The length of each training phase amounts to 2.4M training samples until $64{\times}64$ resolution phase, which lasts for 10.4M samples (totaling in 20.0M). For FFHQ, the $128{\times}128$ phase uses 10.6M samples while CELEBA-HQ and LSUN Cars use 7.5M samples, and LSUN Bedrooms uses 4.8M samples. For FFHQ, the $256{\times}256$ phase uses 5.0M samples, CELEBA-HQ uses 4.5M, LSUN Bedrooms 2.9M samples and LSUN Cars 2M samples. Then training with FFHQ up to $512{\times}512$, this final phase uses 6.7M samples. The training of the final stage was generally cut off when reasonable FID results had been obtained. More training and learning rate optimization would likely improve results. With two NVIDIA Titan V100 GPUs, the training times were 10 days for CELEBA-HQ, 10.5 days for FFHQ $256{\times}256$ and (total) 22.5 days for FFHQ $512{\times}512$, for LSUN Bedrooms 18.5 days, and for Cars 18 days. 3 evaluation runs with different seeds were done for CELEBA-HQ (separately for each configuration of the ablation study, including the full loss with and without layer noise), 3 for FFHQ, 3 for LSUN Bedrooms and 3 for LSUN Cars (1 with and 2 without layer noise). Some runs shared the same pretrained network up to $64{\times}64$ resolution (except in Ablation study, where each run was done from scratch).

For evaluating the model after training, a moving exponential running average of generator weights [18, 24] was used. For visual evaluation, the layer noise can be turned off, often yielding slightly more polished-looking results. For all data sets, training/test set splits were as follows: 60k/10k for FFHQ (download at https://github.com/NVlabs/ffhq-dataset), 27k/3k split for CELEBA-HQ (download with instructions at https://github.com/tkarras/progressive_growing_of_gans), 4,968,695/552,061 for LSUN Cars (download at https://github.com/fyu/lsun), and 3033042/300 for LSUN Bedrooms (download at https://github.com/fyu/lsun). Note that in regular GAN training, complete data sets are often used without train/test split, yielding larger effective training sets. For instance, in FFHQ, we train on the actual 60k training images only, whereas StyleGAN trained on all 70k. For FFHQ and CELEBA-HQ, cropping and alignment of the faces should be performed exactly as described by the authors of the data sets as referred to above, which also direct to the readily available scripts for the alignments (based on facial keypoint detection). For LSUN images, there was no preprocessing except cropping the Cars to $256{\times}256$. Mirror augmentation was used in training the face data sets, but not for training the LSUN data sets (for comparison with prior work).

For baselines in Table 2a and Table 2b, we used pre-trained models for StyleGAN, PGGAN, PIONEER, and GLOW with default settings provided by the authors, except Balanced PIONEER for FFHQ which we trained. FID of PGGAN for Cars and Bedrooms is from Karras et al. [24], whereas FID of FFHQ is from Karras et al. [25] and FID of CELEBA-HQ we computed for $256\times256$ separately. We trained the VAE and WAE models manually. StyleGAN FID for LSUN Bedrooms is from Karras et al. [25] whereas the other FIDs were calculated for $256\times256$ separately. PPLs for StyleGAN and PGGAN for FFHQ come from Karras et al. [25] while the PPL for StyleGAN v2 is from Karras et al. [26], PPL for PGGAN CELEBA-HQ from Heljakka et al. [18] and PPL for CELEBA-HQ of StyleGAN was computed from the pretrained model. For all VAE baselines the weight for KL divergence loss term was 0.005. For all WAE baseline, we used the WAE-MMD algorithm. The weight of the MMD loss term with automodular architecture (WAE-AdaIn) was four and with Balanced PIONEER (WAE-classic) architecture it was two. For VAEs, the learning rate for the encoder was 0.0001, and for the generator 0.0005. For WAEs, the learning rate for both was 0.0002. We trained Balanced PIONEER for FFHQ by otherwise using the CELEBA-HQ hyperparameters, but increasing the length of the $64\times64$ and $128\times128$ pretraining stages proportionally to the larger training set size (60k vs. 27k), training the former up to 20.04M samples and the latter to 27.86M samples, followed by the $256\times256$ stage, which was trained up to 35.4M samples, after which we observed no further improvement. (With shorter pre-training stages, the model training did not converge properly.) Note: Some apparent discrepancies between reported FID results between papers are often explained by different resolutions. In Table 2b we have used $256\times256$ resolution.

For evaluating the encoding and decoding performance, we used 10k unseen test images from the FFHQ data set, cropped the input and reconstruction to $128\times128$ as in Karras et al. [25] and evaluated the LPIPS distance between the inputs and reconstructions. We evaluated 50k random samples in all data sets and compare against the provided training set. The GLOW model has not been shown to work with $256\times256$ resolution on LSUN Bedrooms nor Cars (the authors show qualitative result only for $128\times128$ for Bedrooms).

For Perceptual Path Length (PPL), we repeatedly take a random vector of length $\varepsilon = 10^{-4}$ in the latent space, generate images at its endpoints, crop them around mid-face to $128\times128$ or $64\times64$, and measure the LPIPS between them [25]. PPL equals the scaled expectation of this value (for a sample of 100k vectors).

**Hyperparameter selection** The driving principle to select hyperparameters in this paper was to use the same values as Heljakka et al. [18] whenever necessary, and minimize variation across data sets, so as to show generalization rather than tuning for maximum performance in each data set. The learning rate was attempted at the same rate as in [18] ($\alpha = 0.001$) for the whole length of training. However, the pre-training stages up to $32\times32$ appeared unstable, hence $\alpha = 0.0005$ was attempted and found more stable for those stages. Margin values 0.2, 0.4 and 0.5 were attempted for training stages from $128\times128$ upwards for FFHQ, CELEBA-HQ and LSUN Bedrooms. However, we did not systematically try out all possible combinations, but rather started from the values used in Heljakka et al. [18] and only tried other values if performance seemed insufficient. For the length of the $128\times128$ training stage, separately for each data set, we first tried a long training session (up to 10M seen samples) and observed whether FID values were improving. We selected the cutoff point to $256\times256$ for each data set based on approximately when the FID no longer seemed to improve for the lower resolution. The $256\times256$ phase was then trained until FID no longer seemed to improve, or, in the ablation study, we decided to run for the fixed 40M seen samples. For the $\lambda_\mathcal{X}$ in the image space reconstruction loss, we tried values 1, 0.5 and 0.2, of which 0.2 appeared to best retain the same training dynamics as the L1 loss in CELEBA-HQ, and was hence used for all experiments. Other hyperparameters not mentioned here follow the values and reasoning of Heljakka et al. [18] by default.

# B  Detailed Explanation of Automodulation Architecture

As the structure of the proposed decoder is rather unorthodox (though very similar to Karras et al. [25]) and the modulation step especially is easy to misunderstand, we now explain the workings of the decoder step-by-step in detail.

**Encoder** The encoder works in the same way as any convolutional image encoder, where the last convolution block maps the highest-level features of the input image $x$ into a single 512-dimensional

Input    No smile    Fe/male    Glasses         Input    No smile    Fe/male    Glasses

Scale-
specific →

Regular →

Figure 8: Modifying an attribute in latent space by using only 4 exemplar images of it, for two unseen test images. In 'regular' all-scales manipulation (bottom row), the variance in the exemplars causes unwanted changes in, *e.g.*, texture, face size and pose. When the latent vector only drives the relevant scales, the variance in other scales is inconsequential (top row).

vector $z$. To understand the concept of latent mixing, we can immediately consider having two samples $x_1$ and $x_2$ which are mapped into $z_1$ and $z_2$. In reality, we will use minibatches in the regular way when we train the decoder, but for the purposes of this explanation, let us assume that our batch has only these 2 invidiual samples. Thus our whole latent vector is of size [2, 512]. Each latent vector is independently normalized to unit hypersphere, *i.e.* to reside within [-1, 1].

**Decoder high-level structure** Corresponding to the 7 levels of image resolutions (from 4x4 to 256x256), the decoder comprises of 7 high-level blocks (Fig. 6a). Each such block has an internal structure as depicted in Fig. 6b. In order to understand the modulation itself, the individual activation map of a convolutional layer is the relevant level of abstraction.

**Activation maps** As usual, each deconvolutional operation produces a single activation map per channel, hence for a single deconvolutional layer (of which there are 2 per block), there can be *e.g.* 512 such maps (*i.e.*, 1024 per block). We now proceed to modulate the mean and variance of each of those 512 maps separately, so that each map has two scalar values for that purpose. In other words, there will be 512 + 512 scalars to express the new channel-wise mean and variance for the single decoder layer. As in Eq. (3), the activations of each map are separately scaled to exactly amount to the new mean and variance. Note that those statistics pertain only within each map, *not* across all the maps in that layer.

**Connecting the latent to the scaling factors** In order to drive the modulating scalars with the original 512-dimensional latent vector $z$, we take add a fully connected linear layer that connects the latent vector to each and every modulating scalar, yielding $512 \times 2 \times N$ connections where N is the total number of activation maps in the full decoder (Fig. 6c). Note that this linear layer is not affected at all by the way in which the decoder is structured; it only looks at the latent vector and each convolutional activation map.

**Initiating the data flow with the constant inputs** Hence, given a latent vector, one can start decoding. The inputs to the first deconvolutional operations at the top of the decoder are constant values of 1. This apparently counter-intuitive approach is actually nothing special. Consider that were the latent code simply connected to the first deconvolutional layers with weight 1 to create the mean and with weight 0 for the variance, this would essentially be the same as driving the first layer directly with the latent code, as in the traditional image decoder architecture.

**The data flow** Now, as the image content flows through the decoder blocks, each operation occurs as in regular decoders, except for the modulation step. Whenever an activation function is computed, a separate modulation operation will follow, with unique scaling factors. For the downstream operations, this modulation step is invisible, since it merely scaled the statistics of those activations. Then, the data flow continues in the same way, through each block, until at the last step, the content is mapped to a 3-dimensional grid with the size of the image, *i.e.*, our final generated image.

## C   Scale-specific attribute modification

Interesting attributes of a previously unseen input image can be modified by altering its latent code in the direction of the attribute. The direction can be found by taking N image samples with the attribute and N without it, encoding the images, and taking the difference between the mean encodings. Scaled

(a) FFHQ $512\times512$

(b) CELEBA-HQ $256\times256$

Figure 9: Uncurated random samples for an automodulator trained on FFHQ and CELEBA-HQ, respectively.

as desired, the resulting attribute latent vector can then be added to the latent code of a new unseen input image

The quality of the attribute vector depends on the selected exemplars (and, obviously, on the encoder). Given that all the exemplars have (or, for the opposite case, lack) the attribute A and are randomly drawn from a balanced distribution, then, as N increases (*e.g.*, to $N = 64$), all other feature variation embedded in the latent vector except for the attribute should cancel out. However, for small N (*e.g.*, $N = 4$), this does not happen, and the latent vector will be noisy. For the example in App. B, we now show the difference between applying such a vector on all scales as usually done (*e.g.*, in [18]) in architectures that do not allow latent modulation, and applying it only on the layers that correspond to the scales where we expect the attribute to have an effect (App. B). Here, we simply determine the range of layers manually, as the $16\times16 - 64\times64$ for the smile on/off transform, $8\times8 - 64\times64$ layers for male-to-female, and $4\times4 - 8\times8$ for glasses. The effect of the noise in the attribute-coding vector is greatly reduced, since most of the scales simply are not touched by it.

Note that while autoencoder-like models can directly infer the latents from real exemplar images, in GANs without an encoder, you must take the reverse and more tedious route: the formation of latent vectors needs to take place by picking the desired attribute from randomly generated samples, presuming that it eventually appears in a sufficient number.

## D Random Samples

Our model is capable of fully random sampling by specifying $z \sim \text{N}(\mathbf{0}, \mathbf{I})$ to be drawn from a unit Gaussian. Figs. 9a, 9b and 10 show samples from an automodulator trained with the FFHQ/CELEBA-HQ/LSUN data sets up to resolution $256\times256$.

(a) LSUN Bedrooms

(b) LSUN Cars

Figure 10: Additional samples from an automodulator trained on LSUN Bedrooms and Cars a resolution of at $256{\times}256$.

# E   Reconstructions

We include examples of the reconstruction capabilities of the automodulator at $256{\times}256$ in for uncurated test set samples from the FFHQ and CELEBA-HQ data sets. These examples are provided in Figs. 11 and 12.

Figure 11: Uncurated examples of reconstruction quality in $512\times512$ resolution with unseen images from the FFHQ test set (top row: inputs, bottom row: reconstructions).

Figure 12: Uncurated examples of reconstruction quality in $256\times256$ resolution with unseen images from the CELEBA-HQ test set (top row: inputs, bottom row: reconstructions).

## F  Style Mixing and Interpolation

The well disentangled latent space allows for interpolations between encoded images. We show regular latent space interpolations between the reconstructions of new input images (Fig. 13).

As two more systematic style mixing examples, we include style mixing results based on both FFHQ and LSUN Cars. The source images are unseen real test images, not self-generated images. In Figs. 14 and 15 we show a matrix of cross-mixing either 'coarse' (latent resolutions $4\times4 - 8\times8$) or 'intermediate' ($16\times16 - 32\times32$) latent features. Mixing coarse features results in large-scale changes, such as pose, while the intermediate features drive finer details, such as color.

## G  Comparison to GAN Inversion

Although a GAN trained without an encoder cannot take inputs directly, it is possible to fit images into its latent space by training an encoder after regular GAN training, or by using a separate optimization process. One may wonder how well such image reconstruction would compare to our results here, and we will focus on a readily available method using the latter approach - optimization.

Specifically, we can find the latent codes for StyleGAN [25] with an optimizer, leveraging VGG16 feature projections [38, 42]. The optimization takes place in the large $18\times512$ latent W space, and the resulting latent codes are decoded back to $1024\times1024$ image space in the regular way by the GAN generator network. It should be noted that the latent space of our automodulator is more compact – $1\times512$ – and hence the two approaches are not directly comparable. However, according to Abdal et al. [1], the StyleGAN inversion does not work well if the corresponding original latent Z space of StyleGAN is used instead of the large W space.

Figure 13: Interpolation between random input images from FFHQ test set in $256\times256$ (originals in the corners) which the model has not seen during training. The model captures most of the salient features in the reconstructions and produces smooth interpolations at all points in the traversed space.

Besides the higher dimensionality of the latent space, there are other issues that hamper straighforward comparison. First, the GAN inversion now hinges on a third (ad hoc) deep network, in addition to the GAN generator and discriminator. It is unclear whether inverting a model trained on one specific data set (faces) will work equally well with other data sets. Consider, *e.g.*, the case of microscope imaging. Even though one could apply both the Automodulator and StyleGAN to learn such images in a straight-forward manner as long as they can be approached with convolutions, one is faced with a more complex question about which optimizer should now be chosen for StyleGAN inversion, given the potentially poorer performance of VGG16 features on such images. In any case, we now have a separate optimization problem to solve. This brings us to the second issue, the very slow convergence, which calls for evaluation as a function of optimization time. Third, the relationship of the projected latent coordinates of input images to their hypothetical optimal coordinates is an interesting open question, which we will tentatively address by evaluating the interpolated points between the projected latent coordinates.

First, for the convergence evaluation, we run the projector for a varying number of iterations, up to 200, or 68 seconds per image on average. We use the StyleGAN network pretrained on FFHQ, and compare to Automodulator also trained on FFHQ. We test the results on 1000 CELEBA-HQ test images, on a single NVIDIA Titan V gpu. The script is based on the implementation of Puzer (GitHub user) [38]. To measure the similarity of reconstructed images to the originals, we use the same LPIPS measure as before, with images cropped to $128\times128$ in the middle of the face region. Note that StyleGAN images are matched at $1024\times1024$ scale and then scaled down to $256\times256$ before the cropping. (Note: concurrently to the publishing of this version of the manuscript, an improved version of StyleGAN with possibly better projection capabilities has been released in Karras et al. [26].)

The results (Fig. 16) are calculated for various stages of the optimization for StyleGAN, against the single direct encoding result of Automodulator. The Automodulator uses no separate optimization processes. The results indicate that on this hardware setup, it takes over 10 seconds for the opti-

mization process to reconstruct a single image to match the LPIPS of Automodulator, whereas a single images takes only 0.0079 s for the Automodulator encoder inference (or 0.1271 s for a batch of 16 images). The performance difference is almost at the considerable four orders of magnitude. StyleGAN projection does, however, continue improving to produce significantly better LPIPS, given more optimization. Moreover, to get the best results, we used the $1024 \times 1024$ resolution, which makes the optimization somewhat slower, and has not yet been matched by Automodulator. However, it is clear that a performance difference of $10000\times$ limits the use cases of the GAN projection approach. For instance, in cases where the projected latent codes can be complemented by fast inference, such as Hou et al. [21], the optimization speed is not limiting.

Second, in order to evaluate the properties of the projected latent coordinates, we again projected 1000 CELEBA-HQ test images into the (FFHQ-trained) StyleGAN latent space and then sampled 10000 random points linearly interpolated between them (in the extended W latent space), with each point converted back into a generated image. For comparison, the similar procedure was done for the (FFHQ-trained) Automodulator, using the built-in encoder. We then evaluated the quality and diversity of the results in terms of FID, measured against 10000 CELEBA-HQ training set images. Although such a measure is not ideal when one has only used 1000 (test set) images to begin with, it can be reasonably justified on the basis of the fact that, due to combinatorial explosion, interpolations should cover a relatively diverse set of images that goes far beyond the original images. The results of this experiment yielded FID of $52.88 \pm 0.71$ for StyleGAN and FID of $48.83 \pm 0.95$ for Automodulator. Hence, in this specific measure, StyleGAN performed slightly worse (despite the fact that StyleGAN projection still used nearly 10000x more time).

Third, in order to evaluate the *mixture* properties of the projected latent coordinates, we once again projected 1000 CELEBA-HQ test images into the (FFHQ-trained) StyleGAN latent space, but now take 10000 random pairs of those encodings, and mix each pair in StyleGAN so that the first code drives the first two layers of the decoder, while the second code drives the rest. We then look at the FID against 10000 CELEBA-HQ training set images. We run this for a varying number of iterations (i.e. varying amounts of optimization time) for StyleGAN, and for a direct encoding-and-mixing result of Automodulator. The result (Fig. 17) indicates that the initial StyleGAN projections are inferior to the Automodulator results, but then improve with a larger iteration budget, reaching $26.5\%$ lower FID, but thereafter begin to deteriorate again. Our hypothesis is that by increasing the number of iterations, one finds a StyleGAN latent code that produces a better local projection fit than the earlier iterations (corresponding to a better LPIPS)but resides in a more pathological neighborhood, yielding worse mixing results when combined with another similarly projected latent. More research is needed to investigate this.

Although more research is called for, the FID results suggest that only a fraction of the fidelity and diversity of StyleGAN random samples is retained during projection. More subtle evaluation methods, and *e.g.*, the effect of layer noise, are a topic for future research. For an additional comparison, one could also run similar optimization in Automodulator latent space.

## H    Conditional Sampling

The automodulator directly allows for conditional sampling in the sense of fixing a latent encoding $z_A$, but allowing some of the modulations come from a random encoding $z_B \sim \mathrm{N}(\mathbf{0}, \mathbf{I})$. In Fig. 18, we show conditional sampling of $128 \times 128$ random face images based on 'coarse' (latent resolutions $4 \times 4 - 8 \times 8$) and 'intermediate' ($16 \times 16 - 32 \times 32$) latent features of the fixed input. The input image controls the coarse features (such as head shape, pose, gender) on the top and more fine features (expressions, accessories, eyebrows) on the bottom.

(a) Using 'coarse' (latent resolutions $4\times4 - 8\times8$) latent features from B and the rest from A.

Figure 14: Style mixing of $512\times512$ FFHQ face images. The source images are unseen real test images, not self-generated images. The reconstructions of the input images are shown on the diagonal.

(b) Using the 'intermediate' ($16\times16 - 64\times64$) latent features from B and the rest from A.

Figure 14: Style mixing of $512\times512$ FFHQ face images. The source images are unseen real test images, not self-generated images. The reconstructions of the input images are shown on the diagonal.

**Source B (real)**

(a) Using 'coarse' (latent resolutions $4{\times}4 - 8{\times}8$) latent features from B and the rest from A. Most notably, the B cars drive the car pose.

**Source B (real)**

(b) Using the 'intermediate' ($16{\times}16 - 32{\times}32$) latent features from B and the rest from A.

Figure 15: Style mixing of $256{\times}256$ LSUN Cars. The source images are unseen real test images, not self-generated images. The reconstructions of the input images are shown on the diagonal.

Figure 16: Comparison of LPIPS similarity of image reconstructions in Automodulator (ours) and StyleGAN (left: linear scale, right: log xscale). The error bars indicate standard deviations across evaluation runs. We show that optimization to StyleGAN latent space takes over 3 orders of magnitude more time to match the Automodulator (up to 16 s), but will continue improving thereafter. Here, the Automodulator encodes 1 image in 0.008 s, with the LPIPS shown as the constant horizontal line.

Figure 17: FID comparison of the results of images produced by mixing two reconstructions in Automodulator (ours) and StyleGAN (left: linear scale, right: log xscale), based on three random combination runs. Standard deviations (not visualized) are 0.38 at maximum for StyleGAN and 0.22 for Automodulator. The optimization to StyleGAN latent space takes about 3 orders of magnitude more time to match the Automodulator, continues to improve thereafter, but further optimization of single images leads to worse FID of their mixtures. Here, the Automodulator encodes 1 input image in 0.008 s, with the FID shown as the constant horizontal line.

Figure 18: Conditional sampling of $256 \times 256$ random face images based on 'coarse' (latent resolutions $4 \times 4 - 8 \times 8$) and 'intermediate' ($16 \times 16 - 32 \times 32$) latent features of the fixed unseen test input. The input image controls the coarse features (such as head shape, pose, gender) on the top and more fine features (expressions, accessories, eyebrows) on the bottom.