[Reviews · NeurIPS 2020]

Review 1

Summary and Contributions: The generative autoencoders called automodulators are proposed in this paper. These networks can faithfully reproduce individual real-world input images like regular autoencoders, but also generate a fused sample from an arbitrary combination of several such images, allowing instantaneous ‘style-mixing’ and other new applications.

Strengths: An automodulator decouples the data flow of decoder operations from statistical properties thereof and uses the latent vector to modulate the former by the latter, with a principled approach for mutual disentanglement of decoder layers. Prior work has explored similar decoder architecture with GANs, but their focus has been on random sampling.

Weaknesses: As it is mentioned in the paper, in a typical autoencoder, input images are encoded into a latent space, and the information of the latent variables is then passed through successive layers of decoding until a reconstructed image has been formed. In automodulators, the latent vector independently modulates the statistics of each layer of the decoder so that the output of layer n is no longer solely determined by the input from layer n − 1. The idea of injecting information of the latent space to each layer in the decoder is not new. In BEGAN [1,2], the latent information is fused with the input to each decoder layer. In that paper the authors claim the same conclusion, that this improves significantly the reconstruction quality. The novelty of this paper is then not so strong. I consider that work as an extension of the idea in BEGAN to be able to generate fused samples from an arbitrary combination of several images, allowing instantaneous ‘style-mixing’. [1] Berthelot, D., Schumm, T., Metz, L.: Began: Boundary equilibrium generative adversarial networks. ArXiv abs/1703.10717 (2017) [2] Y. Li, N. Xiao and W. Ouyang, "Improved Boundary Equilibrium Generative Adversarial Networks," in IEEE Access, vol. 6, pp. 11342-11348, 2018. ======================= Post-rebuttal update: Thank the authors very much for detailed responses. I have also read other reviews, and I believe that the authors have exaggerated their claims (due to the BEGAN work), which they have promised to walk back and clarify in the revision. In that case, this work could be marginally above the acceptance threshold.

Correctness: The claims and method are correct.

Clarity: The paper is well written.

Relation to Prior Work: No, important references related to the novelty of the proposed idea are missing.

Reproducibility: Yes

Additional Feedback:


Review 2

Summary and Contributions: This paper introduces a new category of generative autoencoders for learning representations of image data sets, capable of not only reconstructing real-world input images but also of arbitrarily combining their latent codes to generate fused images.

Strengths: This method can faithfully reproduce individual real-world input images like regular autoencoders, but also generate a fused sample from an arbitrary combination of several such images, allowing instantaneous style-mixing and other new applications.

Weaknesses: 1. There are some spelling mistakes in the paper, the author should pay attention to the writing of the paper. 2. Compared with the results of the automatic encoder, the experimental results show that the improvement of this method is very limited.

Correctness: Yes.

Clarity: Generally.

Relation to Prior Work: Not very clear.

Reproducibility: Yes

Additional Feedback:


Review 3

Summary and Contributions: This paper proposes a generative model, with the focus of editing a given real image (for example a style mixing operation), rather than random sampling. The paper uses an encoder and decoder, where the layers in the decoder is “modulated” by the encoder parameters through AdaIN. An orthogonal proposal is a loss that isolates geometric changes to certain parts of the image. The paper evaluates on reconstruction and editing, as well as random sampling. The paper tests on several datasets, including faces bedrooms, and cars.

Strengths: The paper identifies an important issue. While generative models have improved greatly, they are designed for random sampling. One promising application of generative models, such as StyleGAN, is image editing. In that application one must “find" their specific image, and random sampling is of less importance. This paper produces a model aimed at performing editing of an image, by training with an encoder. The method aims to retain properties of StyleGAN, which learns a multiscale disentangled representation, which can be used for style mixing (mixing coarse and fine scales from two images) or random sampling (holding coarse scales fixed and resampling fine layers). The paper produces comparisons to other encoder-decoder methods, generative models, as well as some ablations of itself on various aspects of generative modeling (random sampling with FID and PPL), reconstruction (LPIPS and inference time), and “editing” (FID of mixed images). The paper shows that it can do reconstruction much faster than StyleGAN (Fig 15 in the appendix). The paper tests on several datasets and shows extensive supplemental results.

Weaknesses: While I am excited about the central issue that this paper aims to tackle, I have some questions regarding the execution. To my count, there are 5 losses total. Of these, I understand the (a) reconstruction loss and (b) the scale-specific loss, which was studied in Table 1. I am not convinced that the other losses are necessary/formulated correctly to realize this idea. For example, the code reconstruction loss d_cos is not studied. I have the following questions/concerns about the KL losses: (1) The latent code is normalized to a unit sphere. If latent code is normalized to a unit sphere, why does KL loss to unnormalized N(0,I) still make sense? (2) I do not understand the necessity of the KL loss. Traditionally, the KL loss is used so the latent code can be sampled. Having this likely comes at a cost of the mixing results. If the goal is editing (mixing), why is sampling even necessary? If one wishes to randomly sample, they can use the original stylegan. If one wishes to do mixing, one can use a model focused on just that. I have a difficult time envisioning a scenario where one wishes a single model to do both. (3) The KL gap loss is not explained or ablated. An explanation about why it is asymmetric would also be helpful. (4) Why is KL loss only on encoded reconstructed image and not on original images? I am also unsure why certain losses are in the encoder and certain are in the decoder. For example, the code reconstruction loss d_cos goes from z—>image—>z using the decoder and encoder sequentially. Yet the loss is only applied to the encoder. The generative models are compared to, but just for random sampling. I recommend evaluating them for mixing as well. Figure 15 runs StyleGAN projection, and these results codes can then be mixed. I think this study would bring value in the main paper.

Correctness: I have some concerns about the execution in terms of implementation choices (detailed in weaknesses), but I believe the experiments and methodology are all described accurately and correctly.

Clarity: The abstract and introduction are clear about the purpose of the paper and the related work is thorough. The methods section took me several careful readings and parsings to understand all of the losses. I am likely more familiar with this area than the average reader, who I believe will have a difficult time. So I recommend carefully designing a block diagram to try to make this easier for the reader. Minor issues: * L33 "Previous works on AdaIn are 34 mostly based on GAN models [5, 24] with no built-in encoder for new input images.” AdaIN from [Huang and Belongie 2017] was initially made for style transfer (not GANs!), and was conditioned on an input image * L143 “On” —> “One"

Relation to Prior Work: Yes, the related work is thorough, to my knowledge.

Reproducibility: Yes

Additional Feedback: In summary, I believe the paper identifies an important issue (making generative models useful for editing a specific image), and takes some reasonable steps towards solving the problem. As detailed in the weaknesses section, I am not convinced that this is the simplest and most elegant execution of this idea. But I believe the paper as a whole is informative, and I appreciate the evaluations that the paper shows. ---------- I thank the authors for the explanations in the rebuttal and keep my rating. I look forward to seeing the requested changes.


Review 4

Summary and Contributions: The work introduces a new autoencoder called an Automodulator. Applied to images, this automodulator enables autoencoding single images or performing style transfer and mixing between a group of images, picking and choosing attributes of source images at different scales. The model is based on techniques from the GAN literature (such as adaptive instance norm) and Pioneer Networks. The model operates at multiple scales, enabling transfer of attributes as a specific scale from source images. The authors present a layer-wise cycle consistency technique to encourage disentanglement across the scales of the hierarchy in a self-supervised manner. The authors describe an additional technique to enforce known invariances where multiple samples in a dataset are known to have some shared property in either their coarse or fine properties. The authors present results on FFHQ, CelebA-HQ, LSUN Bedrooms and Cars datasets and claim they achieve high resolution sharp samples comparable to GANs with the representation learning benefits of an encoder. **Update:** Thank you to the authors for the feedback. I will keep my score as it is, but please note my low confidence due to lack of background knowledge.

Strengths: * The model can generate 1024x1024 resolution images, whereas previous work (Pioneer Networks) only worked at lower resolutions. * The samples are reasonably sharp and globally coherent without making use of GAN training. * The style mixing results appear to preserve coarse and fine features from the source images as intended. * The architecture provides an encoder and a procedure for mixing styles between multiple input examples, whereas GAN approaches do not have such an affordance.

Weaknesses: * While the model produces sharper and higher resolution samples than prior autoencoder work, the sample quality and metrics still lag that of implicit generative models such as StyleGAN. * The mixed samples seem prone to unnatural artifacts which may suggest true disentanglement of factors/scales of variation is not achieved. This does not seem to be discussed within the manuscript -- a frank discussion of the failures of the approach and proposals for future work to improve on them would be appreciated. * Scoping the task to "automodulation" (style mixing) seems to avoid the need to study the representation learning power of the architecture. This limits the relevance of the work for further downstream tasks requiring good unsupervised representations and scopes the impact of the work towards creative use cases relying on sample mixing capability.

Correctness: I found no apparent issues with the methodology or claims.

Clarity: I found certain descriptions unclear and in need of multiple read-throughs to understand, but it is not clear whether this is a weakness on the reviewers part of the paper's part.

Relation to Prior Work: The related work section seems robust.

Reproducibility: Yes

Additional Feedback:

[Author Response · NeurIPS 2020]



We thank the reviewers for their insights. We summarize the overall response as positive. The few criticisms presented by R1 and R2 are based on some misunderstandings. We will use page 9 of the camera-ready version to clarify these points and to accommodate the block diagram requested by R3. We address the concerns in the order received.

**R1:** There might be a misunderstanding here. The sole concern of R1 is that our model is an *"extension of the idea in BEGAN to be able to generate fused samples from an arbitrary combination of several images, allowing instantaneous 'style-mixing'."* However, the provided references [1, 2] do not show that BEGAN could do such a thing. We are happy to outline some *fundamental differences to BEGAN*, and will add the same discussion and the citation (in Sec. 2). First, although both models introduce extra paths from latent code to each decoder layer, the similarity ends there. In our model, due to the *modulation*, you can apply latent code $z_1$ at certain layers of the decoder and $z_2 \neq z_1$ at other layers. This enables style-mixing, enforcement of scale-specific invariances, and scale-specific attribute modifications. BEGAN can do none of this, because it connects the one and the same latent code to each of the decoder layers. Unlike in our paper, their skip connections *only* improve the reconstruction quality of the single image. Second, our model (following AGE [43]) consists of nothing but the encoder and decoder, whereas BEGAN *also* contains a discriminator (in your ref. [1], see Sec. 3.5). All else being equal, this makes their model nearly $50\%$ larger.

**R2:** We were puzzled by this review. The criticism of our approach amounts to the claim that *"Compared with the results of the automatic encoder, the experimental results show that the improvement of this method is very limited."* We are not sure what the "the automatic encoder" refers to in this context. If it means 'autoencoder', then we guess this refers to Table 2a. If so, then it is *not at all* the case that the table shows the *"improvement is very limited"*. Table 2a shows our model to be clearly superior to all the baselines, except for the FID of B-Pioneer [17]. Perhaps there was confusion by the fact that, *in Table 2a, the B-Pioneer refers to the classic architecture that cannot do style-mixing* and was only included for reference? In any case, we will clarify this in the table. Our model can do style-mixing and leverage scale-specific invariances and attribute modification. The classic architectures cannot. In the table, you can only compare our model directly against the VAE-AdaIn and WAE-AdaIn.

**R3:** Thank you for good questions and suggestions. We agree that the presented idea could be further improved to make it even more elegant. Yet, without discriminators, our model is already architecturally simpler than a corresponding GAN-based encoder model (such as BEGAN, see above) would be. Regarding comparison to GANs, we will follow your advice and add an experiment evaluating the FIDs of 10,000 mixed latent codes with FFHQ-trained StyleGAN-projections and our model, as a function of optimization time (as in Fig. 15).

**R3** (1): *"If latent code is normalized to a unit sphere, why does KL loss to unnormalized N(0,I) still make sense?"* It turns out one can derive the analytical solution for the KL divergence between an empirical distribution of code vectors normalized to unit hypersphere and the unnormalized unit Gaussian with diagonal covariance. See, *e.g.*, Eq. (7) in [43] or (corrected) Eq. (1) in [16]. You can think of it intuitively as follows: If this loss term is at minimum, where are the points on the unit sphere? They must be uniformly distributed. The larger the loss, the less uniform is the distribution.

**R3** (2): Interestingly, powerful scale-specific editing is not only about encoding the image in the 'appropriate point' in latent space but about the smoothness of the space overall, and the KL divergence minimization encourages the posterior to lie on such a smooth well-behaved manifold. In absence of enforcing this, the path between two latent codes could contain points that decode into low-quality images. This would prevent finding latent vectors that correspond to specific image attributes in latent space (Fig. 4b) or interpolating between images (supplement Fig. 12).

**R3** (3) (and question on $d_{\cos}$): For ablation of the KL gap, see [17]. The code rec. loss ($d_{\cos}$) was studied in [43] (the paper uses L2, but *cos* in the implementation [44]). The code loss is explicitly applied only to the decoder because although the encoder would indeed like to increase it, it already achieves that effect by pushing the codes of generated samples away from Gaussian by the KL terms. For clarity, however, we will summarize these points in the paper.

**R3** (4): The KL loss is also evaluated on original (training) images, see $q_\phi(z \mid x)$ in Eq. (1) and L113 for $x$.

**R4:** Thank you for your encouraging comments! We agree that although a high level of disentanglement is evidently achieved in experiments, certain mixture artifacts indicate that the "disentanglement of factor/scales of variation" is not always perfectly achieved. On page 9, we will add more discussion on this and the prospects for more thorough evaluation. We focused on generalizability and a wide range of experiments rather than exhaustive optimization (such as playing with learning rates and latent space truncation as in StyleGAN [24, 25]).

**R4:** For the general representation learning power of the architecture, we fully agree that measuring performance on auxiliary downstream tasks would be valuable follow-up work. However, we suggest you consider that we do more than *"scope the task to. . . style-mixing"*. After all, we *also* do evaluate reconstruction, random sampling, interpolation (supplement Fig. 12), and attribute modification.

[Meta-Review · NeurIPS 2020]

Generally positive reviews, but please clarify connection to the BEGAN paper, and other reviewer feedback. (They may not have read your reply, since there were problems with the pdf file you uploaded.)